# Clinical Case Report of Non-Diabetic Hypoglycemia Due to a Combination of Germline Mutations in the *MEN1* and *ABCC8* Genes

**DOI:** 10.3390/genes14101952

**Published:** 2023-10-17

**Authors:** Marina Yukina, Ekaterina Solodovnikova, Sergey Popov, Victorya Zakharova, Marina Utkina, Vasiliy Petrov, Ekaterina Troshina, Natalia Mokrysheva

**Affiliations:** State Scientific Center of the Russian Federation Federal State Budgetary Institution, National Medical Research Center of Endocrinology of the Ministry of Health of the Russian Federation, 117036 Moscow, Russia

**Keywords:** non-diabetic hypoglycemia, syndrome of multiple endocrine neoplasia type 1, *MEN1*, congenital hyperinsulinism, *ABCC8*

## Abstract

Introduction: Non-diabetic hypoglycemia (NDH) is a collective term including the multiple causes of hypoglycemic syndrome not due to diabetes mellitus. NDH may result from insulinoma, IGF-2-omas, hypocorticism, Hirata’s disease, genital disorders of glucose metabolism, etc. One of the most common causes of NDH faced by an endocrinologist is insulinoma, which in turn can be part of the hereditary syndrome of multiple endocrine neoplasia type 1 (MEN1). Congenital disorders of glucose metabolism in adult patients, on the contrary, are diagnosed extremely rarely, since they usually manifest in childhood. This article presents a unique clinical case of a patient with NDH and genetically verified MEN1 in combination with congenital hyperinsulinism due to an ABCC8 gene mutation. Case Report: A 43-year-old patient with hypoglycemic symptoms from childhood is presented, in whom multiple pancreatic tumors and fluctuations in glycemia from 38.7 mg/dL to 329.7 mg/dL (2.15 to 18.3 mmol/L) were detected in adulthood, but a mild course of hypoglycemic syndrome was noted. Numerous examinations that were performed to establish an accurate diagnosis are described, signs that served as a reason for expanding the complex of studies are indicated, possible pathogenetic mechanisms of the mild course of hypoglycemic syndrome and hyperglycemic conditions are discussed. Conclusion: This case report is original and highlights that we must always remain intolerant of the inexplicable. Conducting an extended gene study can help perform a correct diagnosis in complex cases.

## 1. Introduction

Hypoglycemic syndrome (HGS) is a condition characterized by hypoglycemia, often accompanied by adrenergic and neuroglycopenic symptoms, which is stopped by the intake of carbohydrates [1]. Hypoglycemia in people without diabetes mellitus (DM) is characterized by a decrease in glycemia of less than 3 mmol/L, developing due to an imbalance in the system maintaining the blood glucose levels. Hyperinsulinemic hypoglycemia—hypoglycemia accompanied by relative hyperinsulinemia (endogenous hyperinsulinism), with the concentration of insulin ≥ 3 U/mL, and that of C-peptide ≥ 0.6 ng/mL—confirms the hyperinsulinemic variant of hypoglycemia [2]. Non-diabetic hypoglycemia (NDH) may result from various diseases and conditions (Table 1).

Insulinoma is the most common cause of non-diabetic hypoglycemia (NDH) in the practice of an endocrinologist and is a type of functional pancreatic neuroendocrine tumor (PNET) that leads to hypoglycemia caused by an inappropriately high secretion of insulin. Based on data from individual countries, the prevalence and incidence of insulinoma among the world population are, on average, 16.4 cases per 1 million population and 0.9 cases per 1 million population per year, respectively [3]. An insulin-producing tumor in 6–8% of cases can be associated with the syndrome of multiple endocrine neoplasia type 1 (MEN1) [4].

MEN1 is a hereditary orphan disease that develops as a result of a germline mutation in the *MEN1* gene. The *MEN1* gene encodes the menin protein, which is a tumor suppressor, the inactivation of which leads to hyperplasia and tumor transformation of various organs and tissues. With this syndrome, combined tumor lesions of the parathyroid glands (PTG) (90%), the gastrointestinal tract (30–70%), including insulinoma, the adenohypophysis (30–40%) develop, besides others [5].

The rarest cause of NDH in adult patients is congenital disorders of glucose metabolism (CDGM). Hypoglycemia in CDGM can be hyper- or hypoinsulinemic. Clinical cases have been described of adult patients with mutations in the *KCNJ1*, *ABCC8*, *GLUD1*, *GCK*, *AGL*, *PGM1*, *ACADM*, *ETFA*, *ETFB*, *ETFDH*, *HMGCL*, *FBP1*, *ALDOB*, *PMM2*, *SLC16A1*, *INSR* and other genes [6]. The diagnosis is confirmed by genetic testing. Genetic testing in adults for CDGM is only reasonable when all other causes of NDH have been excluded. Epidemiological data on the prevalence of CDGM in adults are not available. Most cases of congenital hyperinsulinism in infancy are associated with defects in *ABCC8* and, less often, in *KCNJ1*, and in childhood (up to 16 years) in *GLUD1* [7]. The *KCNJ1* and *ABCC8* genes encode ATP-dependent potassium channels in pancreatic β-cells—Kir6.2 and SUR1, respectively. The disruption of these channels leads to uncontrolled secretion of insulin and to hypoglycemia [8].

Hypoglycemia is dangerous because of the high risk of brain damage up to the development of hypoglycemic coma and death; therefore, timely diagnosis and treatment of the disease are necessary. The disease can also last for a long time in the absence of clear symptoms.

We present a unique clinical case of NDH with a combination of MEN1 with insulinoma and congenital hyperinsulinism caused by a genetic defect in the *ABCC8* gene.

## 2. Case Presentation

A 43-year-old female patient was hospitalized for the first time in the NATIONAL MEDICAL RESEARCH CENTER OF ENDOCRINOLOGY (NMRCE) in January 2017. She complained of bouts of severe weakness and shortness of breath, sometimes preceding fainting, which stopped after a sweet meal, and of weight gain.

A visual inspection of the patient did not reveal clinically significant abnormalities; her height was 172 cm, her body weight was 87 kg, her body mass index was 29.41 kg/m^2^. There were no genetic, endocrine or oncological diseases in the patient’s family. The patient had a history of one birth, and her pregnancy proceeded without complications.

The patient suffered from seizures from the age of seven. She took neurotropic drugs, but this treatment was ineffective. Since the age of 37, attacks had become more frequent. The level plasma glucose level after the usual night fasting was normal. At the age of 42, due to an episode of loss of consciousness, she was hospitalized in a city hospital, where for the first time, a decrease in her glucose level to 50.45 mg/dL (2.8 mmol/L) was registered. She had an MRI of the pancreas, which revealed multiple cystic and solid abnormalities of the head, body and tail. The patient was sent to the NMRCE.

During hospitalization in the NMRCE, we conducted a fasting test. Hypoglycemia (glucose level less than 3 mmol/L) began to be recorded with a glucometer on the second day, but since the level of glycemia was more than 2.2 mmol/L and there were no symptoms (except for the usual weakness), the test continued. Hypoglycemia and neuroglycopenic symptoms were registered only at the end of the test, which lasted 51 h: hyperinsulinemic hypoglycemia was detected (glucose, 38.7 mg/dL (2.15 mmol/L), insulin, 12.97 μIU/mL, C-peptide, 2.03 ng/mL, β-hydroxybutyrate, less than 2.7 mmol/L, substances of oral hypoglycemic drugs, not found). A glucagon test was performed, and the increase in the level of glycemia was less than 1.4 mmol/L. The levels of serum chromogranin A, synaptophysin and antibodies against insulin were within the normal range. Continuous glucose monitoring in the interstitial fluid after fasting showed fluctuations from 61.2 to 329.7 mg/dL (3.4 to 18.3 mmol/L) (Figure 1), but no other criteria indicative of DM were identified. An oral glucose tolerance test was performed with the following results: fasting glucose, 5.95 mmol/L, and 2 h after loading, 6.79 mmol/L; HbA1c 5.5%; HOMA-IR 10. A CT scan of the abdominal organs with contrast confirmed the presence of pancreatic masses (six in total: four cystic, and two solid up to 36 mm—the largest one in the distal part of the tail). The patient underwent selective arterial calcium stimulation with hepatic venous sampling: the release of insulin was approximately the same from all parts of the pancreas. Scintigraphy with octreotide was performed: two foci with overexpression of somatostatin receptors were found in the distal part of the tail of the pancreas (Figure 2). The patient refused a biopsy of the pancreatic tumors.

The patient was diagnosed with primary hyperparathyroidism (PTH levels, 98.9 pg/mL, normal range 15–65), while the levels of albumin-adjusted calcium in the blood were normal, but hypercalciuria (22.44 mg/day, normal range 4.5–14.4) and vitamin D deficiency were detected (20.2 ng/mL, normal range 30–100). Ultrasound of the neck, CT of the neck and mediastinum with contrast, scintigraphy with SPECT were performed, and adenomas of the upper and lower left parathyroid gland were revealed. Densitometry was performed (femoral and radial bones, lumbar spine) as well as gastroscopy, and a decrease in bone density and erosive and ulcerative lesions of the gastrointestinal tract were excluded. MEN1 was clinically diagnosed.

An MRI of the brain was performed, and no pituitary adenoma was detected. According to the results of a laboratory examination, a violation of the secretion of tropic hormones was excluded.

## 3. Treatment

The patient refused the surgical treatment of the pancreatic tumors. The patient was allowed simple carbohydrates only to correct the hypoglycemia and was recommended to consume complex carbohydrates in small portions 4–6 times a day.

Thiazide diuretics, 12.5 mg twice a day, were prescribed to reduce the hypercalciuria. Cholecalciferol was prescribed at 1000 IU per day. Paraadenomectomy of the left upper and lower parathyroid glands was recommended (planned).

## 4. Outcome and Follow-Up

Taking into account the result of the glucagon test, which was uncharacteristic for an insulinoma, as well as the unusually long history of seizures, we assumed that there was some additional disorder affecting the clinical picture of the disease.

An additional blood test was performed and revealed an increase in the level of pancreatic polypeptide to 477.274 pg/mL (normal level, 277) and of GLP1 to 16.3 pmol/L (normal level, 7.7), as well as a decrease in the level of somatostatin to 24.9 pmol/L (normal level, at least 38); chromogranin A, pro-IGF2, GLP2, GIP, CEA, NSE, AFP, β-hCG, glucagon, IGF2, IGF2:IGF1, pro-IGF2/IGF2 were within the normal range.

The molecular genetic testing was performed in the Department of Molecular Genetic Research at the Endocrinology Research Center. Blood samples were obtained from the cubital vein and placed into tubes containing dipotassium (K2) EDTA (ethylenediamine tetra-acetic acid) at a concentration of 1.2–2.0 mg per 1 mL of blood.

Genomic DNA was isolated from peripheral blood lymphocytes using a MagNA Pure-96 robotic station (Roche, Basel, Switzerland) with the MagNA Pure 96 DNA and Viral Nucleic AcidSmall Volume Kits according to the manufacturer’s protocol. A quantitative analysis of isolated DNA was implemented using the Qubit dsDNA BR Assay Kit, the Qubit 2.0 Fluorometer (Invitrogen, Carlsbad, CA, USA) and a fluorescence spectrophotometer (Eppendorf AG, Hamburg, Germany). Genomic libraries were prepared using the KAPA HyperPlus Kit (Roche, Basel, Switzerland), applying the manufacturer’s protocol. The prepared libraries were enriched using the HyperCap Target Enrichment Kit (Roche, Basel, Switzerland) and an Endo custom-designed primer panel.

The custom coding region Endo panel included the following genes: MEN1, VHL, TSC1, TSC2, K-Ras, Yin Yang 1, CDKN2A, MLH1, ADCY1, CACNA2D2/ABCC8, AKT2, EIF2AK3, GATA6, GCG, GCGR, GCK, GLIS3, GLUD1, HADH, HNF1A, HNF1B, HNF4A, INS, INSR, KCNJ11, NEUROD1, PAX4, PDX1, PGM1, PIK3CA, PPARG, PTF1A, RFX6, SLC16A1, UCP2, WFS1. According to the OMIM database, most of them have been associated with hyperinsulinism/insulinoma. The obtained libraries were sequenced by next-generation sequencing (NGS) in paired-end sequencing (2 × 150 bp) with the Illumina Miseq Instrument, using the Miseq Reagent kit v.2 (300 cycles) (Illumina, San Diego, CA, USA).

The sequencing data were processed using an automated algorithm including alignment of reads to the human genome reference sequence (hg19), post-processing of the alignment, identification of variants and filtering of the variants by quality, as well as annotation of the identified variants for all known transcripts of each gene from the RefSeq database using computer algorithms predicting the pathogenicity of the variants, taking into account the recommendations of the American College of Medical Genetics and Genomics (ACMG) and the Russian Institute of Genetics and Genomics (RIMG).

The SpliceAI and AdaBoost programs were used to predict the impact of changes in splice sites and regions of introns adjacent to a splice site. The clinical significance of the identified variants was assessed using OMIM, HGMD and specialized databases for individual diseases (if available) and literature data. The conclusion on the clinical significance of the identified variants was achieved taking into account the recommendations of the American College of Medical Genetics and Genomics (ACMG) and the Russian Guidelines for NGS data interpretation. The conclusion regarded only those variants that were possibly associated with clinical manifestations in the patient. Polymorphisms classified as neutral according to various criteria were not included in the conclusion. Panels with an average depth of coverage of at least 70× and a proportion of target nucleotides with an effective coverage of >10× of at least 97% were analyzed. It should be noted that the NGS method does not provide the reliable identification of insertions and deletions longer than 10 bp, mutations in intronic regions (except for canonical splicing sites), as well as mutations in genes that have a paralog close to a pseudogene sequence. The NGS method is not destined to determine the phase of heterozygous mutation pairs and to detect mutations in the state of mosaicism.

The patient Endo panel had an average depth of coverage of 129× and a width of coverage (10×) of 98%; the genomic assembly version was hg19.

In the *ABCC8* gene (NM 000352.6), a pathogenic single nucleotide substitution was detected in exon 33 in the heterozygous state c.4055G > A, leading to the amino acid substitution p.R1352H (rs28936370; ClinVar: 9098, registration number: VCV000850884.3). This variant has been repeatedly described in the literature as pathogenic in MODY diabetes (PMID: 23563683, 31604004, 31264968, 33046911) with congenital hyperinsulinism (OMIM #240800). A heterozygous pathogenic variant, c.1A > G: p.M1V (rs386134250; HGMD: CM022042; ClinVar: 36525, registration number: VCV000009098.3), was found in the *MEN1* gene (NM 130799.2) in exon 2, described in MEN1 (OMIM #131100).

## 5. Discussion

In the presented clinical case, attention was drawn to the fact that the patient had hypoglycemic symptoms since childhood, but HGS itself was mild (the duration of the fasting test was 51 h), while during the day outside of fasting, the level of glycemia increased to diabetic indicators; a lack of glucose release was found with the glucagon test.

The presence of MEN1 in the patient was not in doubt even before the genetic analysis; so, the first thing that was assumed was a plurihormonal hypersecretion affecting the clinical picture. The simultaneous hypersecretion of various hormones and hormone-like substances in a PNET is a rare phenomenon; in fact, about 6% of cases have been described in the literature [9]. The pancreatic polypeptide is known to have a static effect on insulin and inhibits the release of both glucagon and somatostatin [10]. GLP-1, on the contrary, has insulinotropic and glucagonostatic effects [11]. 

The presence of plurihormonal hypersecretion typical of a PNET in this patient could be the cause of both a mild course of HGS and pronounced fluctuations in glycemia, as well as of a lack of response to the glucagon test.

But the manifestation of hypoglycemic symptoms from childhood did not quite fit into this hypothesis. It is known that approximately half of the patients with NDH are misdiagnosed at the beginning of the disease. Our case was no exception: pre-fainting episodes had been bothering the patient since childhood, which were probably caused by a decrease in glycemia, but the patient had been receiving a pathogenetically unjustified treatment for a long time and in adulthood had clearly adapted to low blood glucose levels. But it is still very doubtful that it was the symptoms of insulinoma that manifested at the age of 7 years. Genetic research solved this mystery.

The detection of the pathogenic mutation in *ABCC8* in adults with nondiabetic hypoglycemia has been described in isolated cases [6], and we did not find descriptions of combinations of germline mutations in *MEN1* and *ABCC8* in the literature. Only in the study by Xiao et al. in 2019, when searching for new somatic mutations in PNET samples, such a combination of genes was detected in a single case. The authors suggested a possible relationship between *ABCC8* and *MEN1*, which requires further research work [12].

Thus, pronounced fluctuations in glycemia can be explained by several mechanisms: the presence of insulinoma, additional plurihormonal hypersecretion, as well as a mutation in ABCC8 with pronounced insulin resistance. The last two mechanisms, we believe, make the greatest contribution to the development of episodes of hyperglycemia. After all, it is known that a mutation in ABCC8 in patients with hypoglycemia in childhood can lead to the development of diabetes mellitus in adulthood [13,14] The alleged mechanism leading to impaired insulin secretion and the development of diabetes mellitus in these patients is considered to involve an inadequate excess intake of Ca++ into β-cells, which leads to their gradual apoptosis [15]. In addition, mutations of this gene have been described in patients with maturity-onset diabetes of the young, i.e., MODY 12. Initially, the differences in phenotypic manifestations are probably related to the activity of KATP channels (overactivity or underactivity) [16]. In the literature, we did not find data on the percentage of cases with the ABCC8 mutation and hypoglycemic syndrome that progress to diabetes mellitus. In this regard, it is necessary to conduct long-term research, collecting information on all such cases, preferably through multicenter international studies.

## 6. Learning Points

The clinical manifestations of insulinoma in MEN1 are variable; it is necessary to take into account a possible plurihormonal hypersecretion.

The manifestation of symptoms of hypoglycemia in childhood in adults, especially in combination with hyperglycemic episodes, is suspicious of CDGM. Conducting an extended gene study can help a correct diagnosis in such complex cases.

## Figures and Tables

**Figure 1 genes-14-01952-f001:**
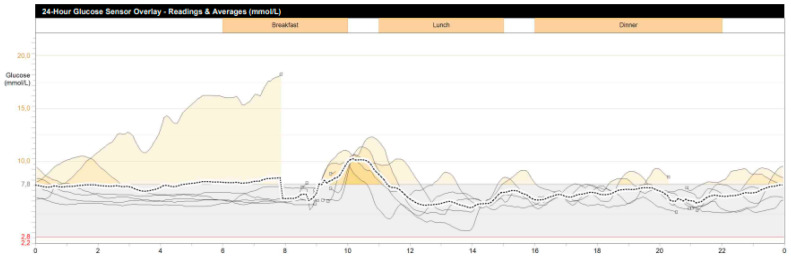
Patient’s continuous glycemic monitoring data for 7 days (Y-axis, glucose (mmol/L), X-axis, time of day; periods of hyperglycemia are indicated in yellow).

**Figure 2 genes-14-01952-f002:**
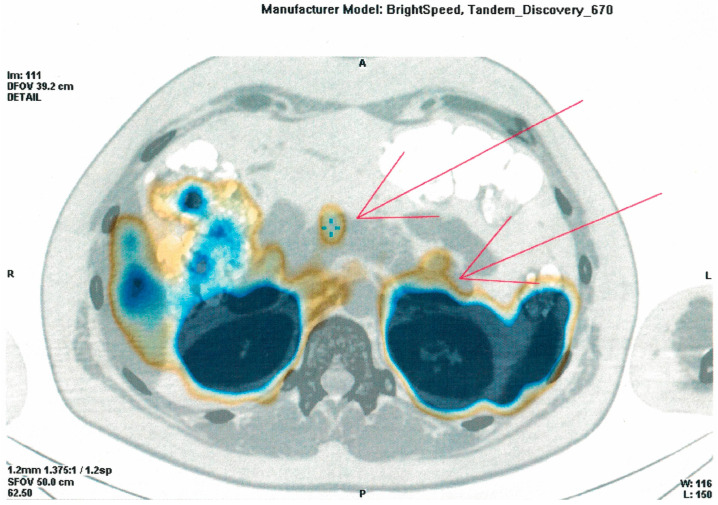
Image of an (111)In-octreotide SPECT/CT scan (arrowheads indicate two foci with overexpression of somatostatin receptors in the distal part of the tail of the pancreas).

**Table 1 genes-14-01952-t001:** Causes of non-diabetic hypoglycemia in adults [2] (modified by the authors).

1. Medicines/products
Alcohol
Quinine-containing products (lemonade, etc.)
Medicines (quinine and its derivatives, indomethacin, glucagon, etc.)
2. Severe diseases
Hepatic, renal or heart failure (in the later stages of the disease)
Sepsis
Cachexia
3. Hormone deficiency
Cortisol
Somatotropic hormone (in children)
4. Non-islet (non-β-cell) tumors with insulin suppression (IGF2-omas)
5. Endogenous hyperinsulinism
Insulinoma
Noninsulinomic pancreatogenic hypoglycemia
Postprandial hyperinsulinemic hypoglycemia due to operations on the upper gastrointestinal tract (late dumping syndrome)
Tumor pathological hypersecretion of glucagon-like peptide-1 and other hormones
Some hereditary disorders of glucose metabolism (hypoglycemia induced by exercise, etc.)
Insulin autoimmune syndrome
Hirata disease
Monoclonal autoimmune hypoglycemia (due to hemoblastosis)
6. Artificial hypoglycemia (insulins and their analogues, sulfonylurea derivatives or repaglinide)
7. Other hereditary disorders of glucose metabolism with insulin suppression

## Data Availability

Data sharing is not applicable to this article, as no data sets were generated or analyzed during the current study.

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
