# Peer review of "Clinical Case Report of Non-Diabetic Hypoglycemia Due to a Combination of Germline Mutations in the MEN1 and ABCC8 Genes"

_genes, 2023, doi:10.3390/genes14101952_

Round 1
Reviewer 1 Report
This case report is original and highlights that we must always remain intolerant of the inexplicable! This made it possible to make the diagnosis of ABCC8 in adulthood. But the writing of the article must be more precise and rigorous.
1-Abstract and Introduction: the authors announce multiple causes of non-diabetic hypoglycaemia (NDH) but only cite some of them. I would suggest mentioning that certain hypoglycemias arise from a sometimes obvious context (Reactive postprandial, iatrogenic, Alcohol,Hepatic, Cardiac, Renal failure , Severe sepsis, anorexia, cachexia, Cortisol, GH insufficiency, Surreptitious, bariatric surgery.......).
2-Line : 38 : only neurological ? not adrenergic ? Specify why.
3- Biological definition of Hypoglycaemia ? of endogenous hyperinsulinism ?
4-Line 40 : I do not agree with the statement "Insulinoma is the most common cause of NDH!!! It is a very rare cause!! Moreover, specify the prevalence outside the situation of a MEN 1.
5-Line 51 : I think it is not accurate to equate CDGM only to the causes of congenital hyperinsulinism...even in adults : put the sentence line 56 "hypoglycaemia in ...." at the beginning of this paragraph.
6-Line 57: "most cases...." specify : in childhood? in adulthood ? ...and please rely on several references: congenital infancy-onset hyperinsulinism = most common mutation ABBC8, childhood-onset hyperinsulinism = most common = GLUD1, congenital Adult-diagnosed hyperinsulinsim: most common ?
See Hopkins JJ et al, JCEM 2023;
7- Line 53 : Why don't the authors use why don't the authors use the usual term"congenital hyperinsulinism"
8-Line 61 : At what threshold is hypoglycemia dangerous? What is the neurological risk?
9-Line 64, you need to be clearer : MEN1 with insulinoma
10-Line 71 : did she have gain weight ? as it is usual with insulinoma and some of the patients with congenital hyperinsulinism.
11- line 82 : you mean at 72 h ? not after ?
12-what were the blood glucose level at 66 or 68 hours?
What is your diagnostic threshold for hypoglycemia? re-specify the criteria for endogenous hyperinsulinism. Was the search for hypoglycemic drugs negative?
13-Can you specify the values of the oral glucose tolerance test?
14-Can you show the data from the octreotide scintigraphy?
15 - Vitamin D value ?
16-has the patient had any pregnancies? and if so, was there diabetes or hypoglycemia?
17-Is postprandial hyperglycemia explained by the inhibition of normal beta function by insulinoma or by the plurihormonal hypersecretion or by the ABCC8 mutation or by all theses mecanisms together ? can you clarify your main hypothesis?
18-specify in the discussion the usual percentage of patients with ABCC8 progressing to diabetes.
Author Response
Thank you for your comments.
We have carefully reviewed the comments and have revised the manuscript accordingly. Our responses are given in a point-by-point manner in the article. Changes to the manuscript are shown in yellow. We hope the revised version is now suitable for publication and look forward to hearing from
you in due course.
Enclosed you will find a copy of the article with comments and corrections.

Reviewer 2 Report
This paper reported a clinical case of Non-diabetic hypoglycaemia duo to ABCC8 mutation, combined with MEN1. Overall, it's an interesting case, and the authors' description was concise and clear.
The only obvious issue is the English language. I highly recommend the authors to seek for help from a professional English writing personnel, or a native speaker, to polish the language to meet the publication requirement.
The whole article needs English language improvement.
Author Response

(The authors gave the same response as above.)
